# Naïve individuals promote collective exploration in homing pigeons

Gabriele Valentini[1,2]*, Theodore P Pavlic[2,3,4,5,6,7], Sara Imari Walker[1,4,8], Stephen C Pratt[3], Dora Biro[9,10], Takao Sasaki[11]

[1]Arizona State University, School of Earth and Space Exploration, Tempe, United States; [2]Arizona State University, School of Life Sciences, Tempe, United States; [3]Arizona State University, Beyond Center for Fundamental Concepts in Science, Tempe, United States; [4]Arizona State University, School of Computing and Augmented Intelligence, Tempe, United States; [5]Arizona State University, School of Sustainability, Athens, United States; [6]Arizona State University, School of Complex Adaptive Systems, Tempe, United States; [7]Arizona State University, ASU–SFI Center for Biosocial Complex Systems, Tempe, United States; [8]Santa Fe Institute, Santa Fe, United States; [9]University of Oxford, Department of Zoology, Oxford, United States; [10]University of Rochester, Department of Brain and Cognitive Sciences, Rochester, United States; [11]University of Georgia, Odum School of Ecology, Athens, United States

**Abstract** Group-living animals that rely on stable foraging or migratory routes can develop behavioural traditions to pass route information down to inexperienced individuals. Striking a balance between exploitation of social information and exploration for better alternatives is essential to prevent the spread of maladaptive traditions. We investigated this balance during cumulative route development in the homing pigeon *Columba livia*. We quantified information transfer within pairs of birds in a transmission-chain experiment and determined how birds with different levels of experience contributed to the exploration–exploitation trade-off. Newly introduced naïve individuals were initially more likely to initiate exploration than experienced birds, but the pair soon settled into a pattern of alternating leadership with both birds contributing equally. Experimental pairs showed an oscillating pattern of exploration over generations that might facilitate the discovery of more efficient routes. Our results introduce a new perspective on the roles of leadership and information pooling in the context of collective learning.

*For correspondence:
gabriele.valentini.85@gmail.com

Competing interest: The authors declare that no competing interests exist.

## Editor's evaluation

This study in the field of collective behavior addresses how naïve and experienced individuals (i.e., homing pigeons) pool information in order to navigate while flying back home. The authors show that the passage of information is largely democratic, meaning information passes both ways, and that, unexpectedly, exploration of the route is initiated both by naïve and experienced birds. The work provides a new perspective on information sharing during collective learning.

## Introduction

The coordinated motion of groups is a widespread phenomenon observed in multiple taxa (*Vicsek and Zafeiris, 2012*). Among other adaptive advantages, such as increased energetic efficiency and decreased odds of predation (*Krause and Ruxton, 2002*), collective motion also allows group members to increase their sensory and cognitive capacity (*Berdahl et al., 2013*; *Gelblum et al.,*

*2020*) and to acquire valuable social information for navigation (*Couzin, 2009*; *Couzin et al., 2011*). In many animals, this social information concerns well-established foraging or migratory routes that can, in some species, persist over successive generations (*Helfman and Schultz, 1984*; *Sasaki and Biro, 2017*; *Jesmer et al., 2018*). Knowledge and skills that accumulate over generations can provide groups with an enhanced ability to solve difficult problems (*Biro et al., 2016*). Not only can later generations build on the success of earlier ones, but the introduction of new members, even those with no prior knowledge, adds diversity that can enhance the group's behavioural solutions (*Mehlhorn et al., 2015*). As is often the case (*Hills et al., 2015*), behavioural patterns that lead to a search for improvement, whether individually, socially, or over multiple generations, involve an exploration–exploitation trade-off. In navigation problems, both solitary individuals and groups have to balance between exploiting previously acquired information necessary to navigate a known route and exploring for additional information that might allow them to approach the optimal route (*Fu and Gray, 2006*). However, how moving collectives compromise between these tasks has received limited attention.

Understanding the exploration–exploitation trade-off is complicated by ambiguity about group leadership (*Couzin et al., 2005*; *Garland et al., 2018*). Although some collectives (e.g., ants, honeybees) can allocate certain individuals to spatial exploration while others continue to exploit accumulated information (*Hills et al., 2015*), individuals in cohesively moving groups are highly coupled and can only benefit from compromising between exploring and exploiting if they do so in unison. For example, dominant guineafowls displace subordinates to monopolize a foraging patch (i.e., exploitation) but, to benefit from the safety of group cohesion, are then forced to follow subordinates in their exploration for alternative patches (*Papageorgiou and Farine, 2020*). Baboons can also compromise between movement decisions. When they travel together, they follow one member's directional preference over another if the angle of disagreement between conflicting preferred directions is large but compromise by averaging alternative proposed directions when this angle is small (*Strandburg-Peshkin et al., 2015*). If individuals are to stay together, the group must reach consensus between following a known route or departing from it to find better routes, foraging patches, or temporary resting locations. Elucidating whether different group members contribute differently to this process is crucial to understanding how groups compromise between exploration and exploitation.

We investigate this question in the context of navigation through natural landscapes using the homing pigeon *Columba livia* as our model system. After successive homing journeys from a given release site, pigeons develop stable idiosyncratic routes that are followed with high fidelity (*Meade et al., 2005*; *Guilford and Biro, 2014*). These birds rely on sequences of localized visual landmarks to recapitulate familiar yet individually distinct routes (*Biro et al., 2004*; *Meade et al., 2005*). Each route is learned in a gradual process starting with an exploration phase that samples new landmarks, from which the bird eventually converges upon a stable sequence of landmarks (*Biro et al., 2004*). Experiments with paired birds show that route information can be passed from experienced birds to naïve individuals through social learning (*Pettit et al., 2013a*) and can be modified through information pooling when individuals with different idiosyncratic routes share information to reach a compromise between their routes (*Biro et al., 2006*). Although learning generally improves route efficiency, both social learning and information pooling tend to reach a plateau beyond which further improvement in efficiency is not seen. However, for birds flying together, the introduction of a naïve individual in place of an experienced one effectively leads to the resumption of exploratory behaviour and further route improvement (*Sasaki and Biro, 2017*). Yet, it remains unclear to what extent a bird's prior experience influences the balance between exploration and exploitation and how birds with potentially different route preferences jointly shape a route.

Indeed, the mechanisms underlying how different individual preferences are combined into a collective outcome is one of the key foci in studies of collective animal behaviour. Broadly, group decisions can range from despotic with a single leader to democratic in which input from different individuals is aggregated to reach consensus (*Conradt and Roper, 2003*). Some animal groups make both despotic and democratic decisions, and researchers have been investigating what determines reliance on one collective decision-making strategy over the other (*King and Cowlishaw, 2014*). For example, baboons live in despotic societies where the alpha male is most often responsible for group decisions between alternative foraging destinations (*King et al., 2008*), but they can also decide democratically in certain situations, such as during daily ranging activities (*Strandburg-Peshkin et al., 2015*). Evidence of both despotic and democratic decisions also exists in homing pigeons (*Biro et al.,*

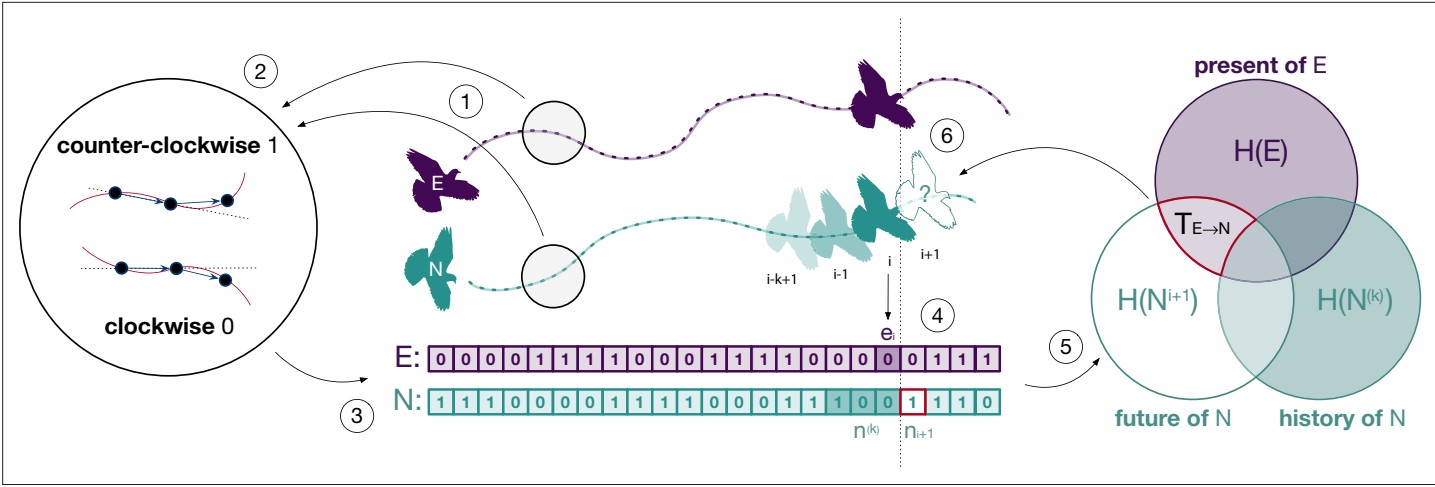

**Figure 1.** Illustration of the methodological approach. The spatial trajectories of an experienced (E) and a naïve (N) bird (point 1) are encoded as clockwise and counterclockwise rotations (point 2) which we represent as discrete time series (point 3). The combination of rotations encoded in both series (point 4) is used to estimate the probabilities required to compute transfer entropy (point 5) and to determine the influence of one individual over the future behaviour of the other (point 6). This example illustrates transfer entropy from experienced to naïve, but we also computed it for the opposite direction.

*2006*; *Nagy et al., 2010*; *Jorge and Marques, 2012*). When leadership is defined as disproportionate input into collective navigational decisions, either through spatial position (*Pettit et al., 2013b*), route similarity (*Flack et al., 2012*), or directional correlation delay (*Nagy et al., 2010*), a number of different factors have been shown to play a role in it. Leadership dynamics are influenced by individual differences among birds in fidelity to their own routes (*Freeman et al., 2011*), their typical flight speed (*Pettit et al., 2015*), their personality (*Sasaki et al., 2018*), as well as their level of experience (*Flack et al., 2012*). Moreover, equally experienced birds are known to come to a compromise by averaging their idiosyncratic routes so long as the pair's route remains within a threshold distance from each bird's favoured one – a low level of conflict. Higher levels of conflict lead instead to a splitting of the pair or to the emergence of a single leader (*Biro et al., 2006*). Nonetheless, experience alone is unable to fully recover the leadership structure characteristic of larger flocks (*Watts et al., 2016*). Spatial position offers some insight into leadership; on average, birds flying closer to the front of the flock have a stronger influence on the flock's directional choices than birds flying at the back (*Nagy et al., 2010*; *Pettit et al., 2013b*). Even so, the moment-to-moment relationship between leadership and level of experience remains unclear.

Leader–follower interactions of this sort can be accurately captured using information-theoretic measures that quantify causal relations in terms of predictive information (*Butail et al., 2016*; *Kim et al., 2018*; *Crosato et al., 2018*; *Ray et al., 2019*; *Valentini et al., 2020*). This methodological approach, which generally requires large amounts of data (but see *Porfiri and Ruiz Marín, 2020*), is gaining popularity among behavioural ecologists (*Strandburg-Peshkin et al., 2018*; *Pilkiewicz et al., 2020*) as tools for automatic monitoring and extraction of the necessary volumes of behavioural data become increasingly available (*Egnor and Branson, 2016*). One of these measures, *transfer entropy*, quantifies information about the future behaviour of a focal individual that can be obtained exclusively from knowledge of the present behaviour of another subject (*Schreiber, 2000*). Transfer entropy measures information transferred from the present of the sender to the future of the receiver (*Lizier and Prokopenko, 2010*). It explicitly accounts for autocorrelations characteristic of individual birds' trajectories (*Mitchell et al., 2019*) by discounting predictive information available from the sender's present that is already included in the receiver's past (see *Figure 1*). Furthermore, it does not require a model of how sender and receiver interact, and it is well suited to study social interactions both over space and time (*Lizier et al., 2008*; *Strandburg-Peshkin et al., 2018*). This aspect of transfer entropy encompasses traditional methods to quantify collective movement that are based on modelling an individual's behaviour as a combination of three motional tendencies (*Couzin et al., 2002*) – alignment of direction to nearby group members, attraction towards sufficiently distant members, and repulsion from sufficiently close members – that allow an individual to maintain proximity to the

group. In this context, transfer entropy is advantageous as it can capture causal interactions due not only to alignment forces (*Nagy et al., 2010*) but also to attraction and repulsion forces that result in temporarily unaligned states (*Pettit et al., 2013b*).

We study collective decision-making and the exploration–exploitation trade-off using an experimental analysis of cumulative route development in homing pigeons conducted by *Sasaki and Biro, 2017*. In their experiments, pairs consisting of a naïve and an experienced bird were required to successively solve the same homing task a total of 12 times. This set of paired flights, which represents a single generation of cumulative route development, allowed the naïve bird to acquire knowledge of localized visual landmarks necessary for homing. At the end of each generation, the more experienced bird was then replaced with a new naïve individual and the learning process was repeated with the newly formed pair. This transmission-chain design, where experienced individuals were repeatedly replaced with naïve ones, lasted five generations and was replicated in 10 independent transmission chains. Route efficiency was measured as the ratio of the beeline distance between the release site and the home loft (i.e., the ideal optimum) and the actual distance travelled by birds. Sasaki and Biro's (2017) s results showed that, although homing efficiency dropped considerably every time a new naïve bird was introduced, transmission-chain pairs continued to improve within and over generations, eventually outperforming both solo and fixed-pair controls (respectively, 0.92 efficiency versus 0.83 and 0.85). In contrast, the efficiency of solo and fixed pairs plateaued after they had first established their idiosyncratic routes (around the 9th–10th release for the former and the 7th–8th release for the latter).

The continued improvement seen in transmission chains might result from a variety of decision-making mechanisms ranging from fully despotic to increasingly democratic. A simplified perspective of this continuum allows us to consider four alternative hypotheses. In two of these alternatives, $H_1$ and $H_2$, and, a single despotic leader, either the naïve ($H_1$) or the experienced ($H_2$) bird, determines the entire homing route. Whereas evidence of social learning (*Sasaki and Biro, 2017*) suffices to dismiss the possibility of leadership by the naïve individual ($H_1$), leadership by the experienced individual ($H_2$) could still be the only process in place if social learning is unidirectional and the naïve individual merely triggers the experienced bird to resume and lead exploration. Under the other two hypotheses, $H_3$ and $H_4$, birds pool their personal information by means of democratic processes based on moment-by-moment integration of individual preferences or transient, alternating leadership (*Conradt, 2012*). The third hypothesis ($H_3$) entails the experienced bird contributing only its past route information and relying instead on the naïve individual for the discovery of route innovations. If this hypothesis holds, we expect the naïve bird to disproportionally lead phases of exploration. Otherwise (fourth hypothesis, $H_4$), both experienced and naïve birds might contribute through exploration to the discovery of new information.

We discriminated between these alternative hypotheses by using transfer entropy to reveal the extent to which birds influence each other and to investigate if relative spatial position can accurately predict leader–follower dynamics. On this basis, we studied the contribution of each bird to the exploration–exploitation trade-off over different stages of route development. This exploration–exploitation perspective of homing route development allowed us to characterize the efficiency of choices made by birds over the course of the experiment and to shed light on the superior performance of experimental pairs with respect to solo and fixed-pairs controls.

## Results
### Birds pool information

Whereas previous evidence of social learning (*Sasaki and Biro, 2017*) suffices to dismiss the possibility of naïve individuals behaving in a despotic manner ($H_1$), the despotic approach remains a possible option for experienced birds ($H_2$). Indeed, the social learning hypothesis under which the naïve bird passively copies the idiosyncratic route of the experienced one (i.e., $H_2$, the despotic leader) entails a transfer of information that is unidirectional – from the experienced to the naïve bird. Instead, under the two alternative hypotheses based on democratic decision-making ($H_3$ and $H_4$), the two birds rely on bidirectional information transfer to pool information and increase the efficiency of their route (*Pettit et al., 2013a*; *Sasaki and Biro, 2017*). We rejected the unidirectional social learning hypothesis $H_2$ by finding causal evidence of information pooling; the naïve bird actively influenced the behaviour

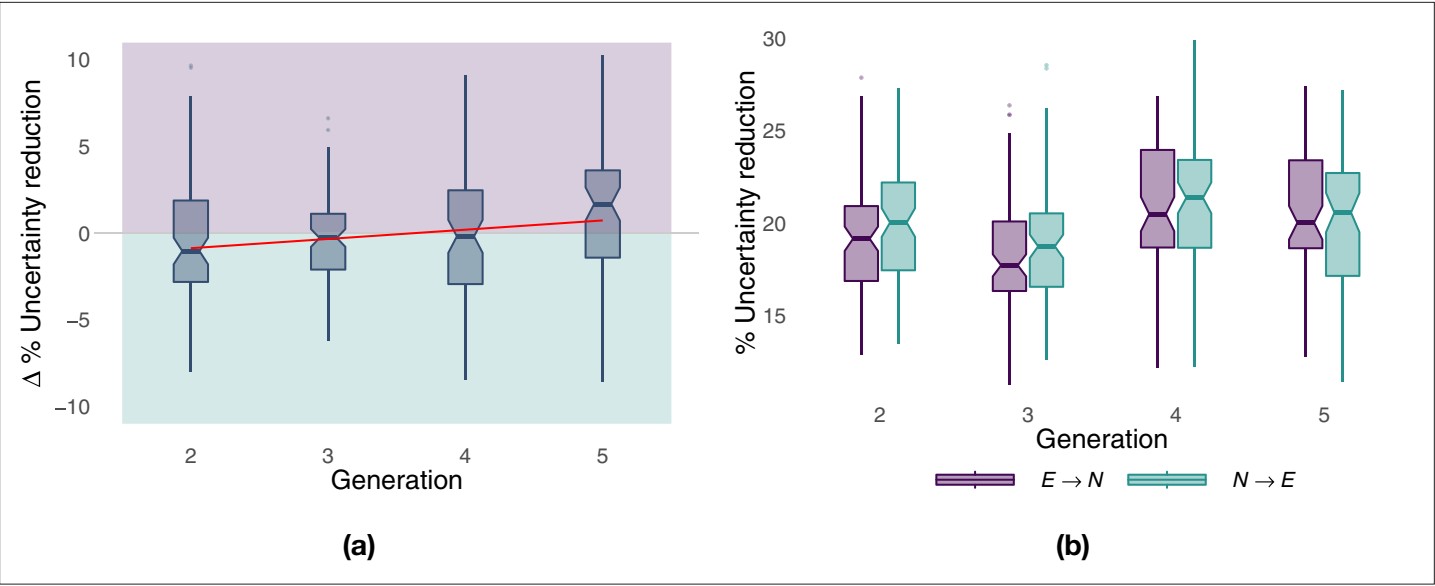

**Figure 2.** Predictive power of birds over generations. Panel (**a**) shows the net predictive power of the two birds over generations; it measures the excess predictive information within the pair and highlights which of the two birds is more informative (purple for the experienced bird, green for the naïve bird). The red line corresponds to a linear fit over generations using the Theil–Sen estimator. Panel (**b**) shows the predictive power of naïve and experienced birds separately from each other as a function of generations. The predictive power of a bird with respect to the other is measured in terms of the percentage reduction in uncertainty, and it has been computed on the basis of transfer entropy as detailed in Materials and methods.

of the experienced one for a large portion of the parameter space (**Appendix 1—figure 1**). As is common practice with these measures (**Porfiri, 2018**), we selected the parameter configuration that maximized the total transfer of information between the two birds (one sample every 0.2 s, history length of 10 samples). This was maximal for the shortest sampling period (i.e., prediction interval) of 0.2 s and progressively decreased towards 0 for larger periods up to 4 s, indicating that the effect of an interaction between birds was transitory and lasted for a limited period of time. Using this configuration, we compared measurements of information transfer against those of a surrogate dataset created by pairing trajectories of birds that were not flown together. We found that levels of mutual influence between birds that flew together were significantly higher than those observed in the surrogate dataset both overall and for each generation separately (Mann–Whitney–Wilcoxon, columns 2 and 3 of **Appendix 1—table 1**).

During the first two generations of paired flights (**Figure 2a**, paired analysis), when there was a large margin to improve the efficiency of the pair's trajectory, the naïve bird was more informative than the experienced one, evidenced by a stronger influence of the former over the latter. At generation 4, there was a balance between the two birds, whereas the experienced bird eventually became the better source of predictive information in the last generation. A linear fit over generations of the paired comparison (**Figure 2a**, red line) showed an increasing influence of the experienced bird over the naïve one (Theil–Sen estimator, slope 0.534, p<0.001). Additionally, a non-paired comparison of the same results revealed that, although the behaviour of the naïve bird resulted, on average, in a marginally higher predictive power than that of the experienced bird (18.7–21.4% versus 17.7–20.5%), variations in each bird's route explained a large portion of the other bird's behaviour (**Figure 2b**), suggesting non-trivial leadership dynamics. These results do not show whether different levels of experience within the pair led to asymmetric contributions of birds to route development, with the experienced bird providing only its past route information and the naïve bird in charge of discovering route innovations, or if both birds contributed to the exploration for possible route alternatives. To discriminate between the remaining hypotheses $H_3$ and $H_4$, we first developed the means to evaluate leadership on a moment-to-moment basis.

## Relative position determines temporary leadership

Consistent with information sharing within each pair, we found that experienced and naïve birds repeatedly switched their positions at the front and back of the pair (**Figure 3a**). Previous studies

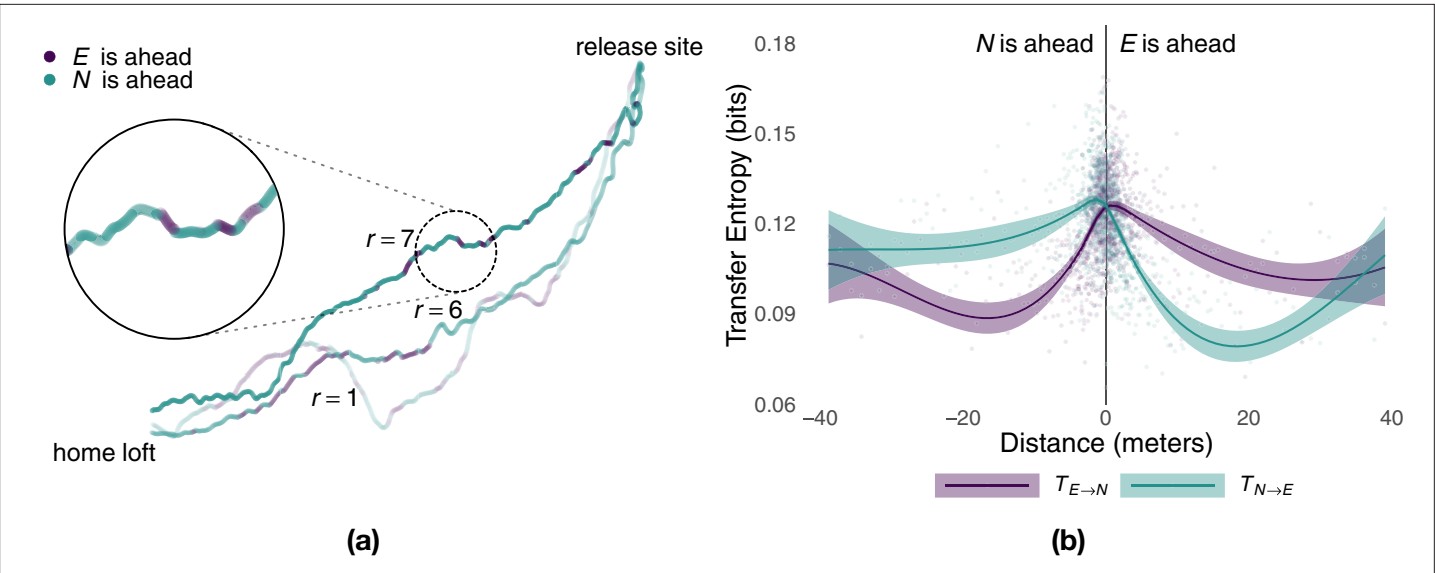

**Figure 3.** Predictive power of birds as a function of relative distance. Panel (**a**) shows sample flight trajectories for a number of different releases, *r*, of the same pair of birds. Colours highlight which bird is ahead of the other during different segments of the route. Panel (**b**) shows the local transfer of information (mean and 95% confidence interval) between the experienced bird and the naïve one as a function of their relative distance (colours represent the direction of information transfer) estimated from generations 2–5 using smoothed conditional means.

found evidence that birds that spent, on average, more time at the front of the flock had a tendency to assume leadership roles (*Nagy et al., 2010*; *Pettit et al., 2013b*). To see whether this average relationship between leadership and position holds at each point in time, we investigated the spatio-temporal dynamics of information transfer (*Lizier et al., 2008*). We did so by considering the amount of predictive information obtained by each bird as a function of the distance from the experienced to the naïve bird projected over their mean direction of motion (*Figure 3b*). We found that within a distance of up to 30 m the bird flying ahead was consistently more informative than that flying in the back. This is not only further evidence that the bird flying ahead acts as the leader, influencing the path of the follower behind it, but, because of its finer grain, it also enables relative distance between birds to be used as a (more parsimonious) moment-to-moment measure of causal influence within the pair.

The experienced and the naïve birds alternated leading segments of the route, where leadership durations were consistent with a log-normal distribution (see Supplementary material). Although the naïve bird flew at the front of the pair for longer segments (Whitney–Mann–Wilcoxon, p<0.047, *W* = 56865638), the difference was very small (0.3 s) and largely driven by the flights of one generation. Indeed, for all generations but the third (p<0.001, *W* = 6588674), the distribution of consecutive time spent at the front of the pair by the experienced bird cannot be distinguished from that of the naïve individual (*Appendix 1—table 2*). The tails of these distributions approach that of an exponential distribution and suggest that temporary leadership might be decided randomly (*Biro et al., 2006*) instead of using deterministic rules such as fixed periods of time. Moreover, with the exception of generation 3 where 54% of the route was led by the naïve bird (Wilcoxon signed-rank test, p=0.03, *V* = 1851), there was no significant difference in the proportion of a flight spent by each bird at the front of the pair (*Appendix 1—table 3*), suggesting a relatively egalitarian relation between birds despite differing levels of experience.

## Exploration–exploitation dynamics explain flight performance

*Sasaki and Biro, 2017* previously showed that flight efficiency varies across treatments with experimental pairs eventually outperforming both fixed pairs of birds and solo individuals. The discovery of route innovations and, in particular, how birds with different levels of experience contribute to this task, is the key to the superior performance of experimental pairs. To understand this phenomenon and thus shed light on the pair's information-pooling mechanism, we investigated how pigeons balance between exploitation of known information – closely following (<300 m) their most recent

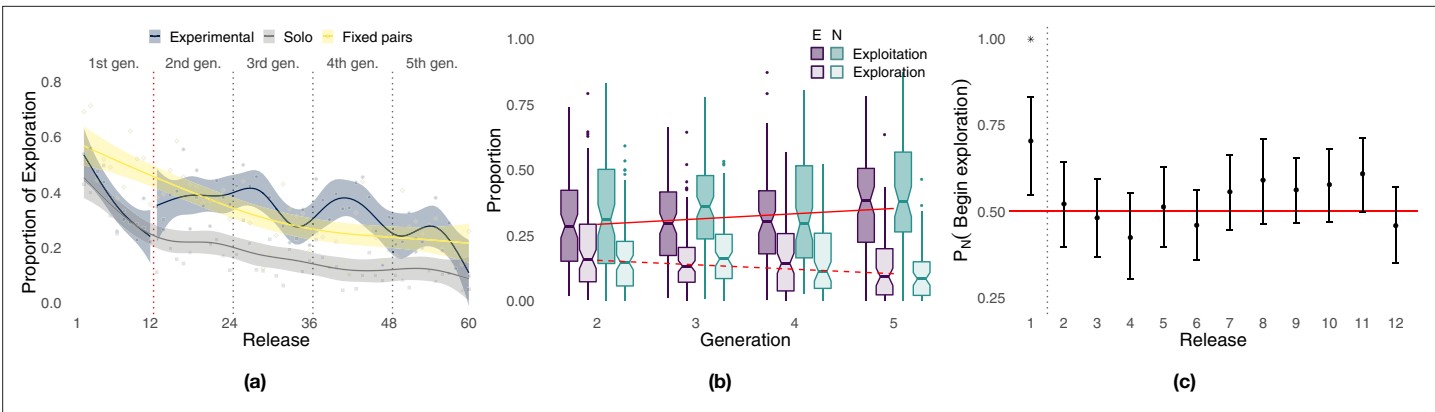

**Figure 4.** Analysis of exploration and exploitation. Panel (**a**) shows the proportion of exploration over releases for the experimental group (the red dotted vertical line separates solo flights at generation 1 from paired flights at generations 2–5), the solo control, and the fixed-pairs control. Smoothed lines are computed using generalized additive models using shrinkage cubic regression splines (mean and standard error); points represent averages for individual releases. Panel (**b**) shows the proportion of a flight led by each bird during phases of exploration and exploitation in the experimental treatment. Darker colours correspond to exploitation, lighter colours to exploration; purple represents the experienced bird, green the naïve one; red lines represent linear fits to data pooled from both birds using the Theil–Sen estimator (slopes and p-values: respectively, 0.0203, p<0.001 for exploitation and –0.0171, p=0 for exploration). Panel (**c**) shows the probability for the naïve individual to initiate phases of exploration over releases (exact binomial test, p<0.01 for release 1). The dotted vertical line separates the first release, where the naïve bird was significantly more likely to initiate, from the successive releases 2–12 showing no significant difference between the two birds; the red line provides a visual reference indicating an equal likelihood between the two birds to initiate exploration.

route – and exploration for possible route improvements. To do so, we labelled segments of flight trajectories as a function of the point-to-point distance from each point of a focal trajectory to the closest point of the immediately preceding trajectory (i.e., baseline) and compared the exploration–exploitation dynamics both across treatments and between experienced and naïve birds.

During the initial part of the experiment (*Figure 4a*, first 12 releases), exploration decreased steadily in all conditions with birds that flew individually in the experimental group (i.e., generation 1) performing similarly to those of the solo control (respectively, 36.7 and 34.2%), whereas fixed pairs of birds explored significantly more (51.7%, Whitney–Mann–Wilcoxon, p<0.001, *Appendix 1—table 4*). However, while exploration steadily decreased for solo and fixed pairs of birds in the successive 48 releases, experimental pairs showed a markedly different pattern of exploration oscillating over generations (*Figure 4a*, releases 13–60). Each time a new naïve individual was paired with an experienced one (dotted vertical lines), exploration increased for about 5–6 releases, reaching values well beyond those of both solo birds and fixed pairs; then exploration decreased within a few releases (2–4) to the same levels as those of fixed pairs. On average during generations 2–5, experimental pairs explored (32.9%) significantly more than both solo (15.7%, p<0.001) and fixed pairs of birds (29.3%, p=0.0456). These results also held when exploration and exploitation were defined with respect to the last release of the previous generation. Under this model, differences between experimental and fixed pairs were even more pronounced, with the former characterized by 46.6% exploration and the latter by only 32.4% (p<0.001, *Appendix 1—table 5* and *Appendix 1—figure 6*). The inferior flight efficiency of solo and fixed pairs of birds might thus be explained, at least in some measure, by a lower likelihood to discover route improvements due to limited exploration.

The superior homing performance of experimental pairs is suggested to be rooted in their ability to select novel portions of a route introduced by the naïve individual that are more efficient while discarding inefficient ones (*Sasaki and Biro, 2017*). Under this hypothesis, which is an extension of $H_3$, we expect to observe not only increasing homing efficiency over generations but also an asymmetric pattern of leadership in which the naïve individual leads periods of exploration and the experienced one leads periods of exploitation. We found instead no significant difference between the contributions of the experienced bird and those of the naïve one both overall and within generations (*Figure 4b*, *Appendix 1—table 1*). The sole exception is represented by generation 3 during which the naïve bird contributed more than the experienced one to exploitation (Wilcoxon signed-rank test, p=0.035, V = 1871). In agreement with our expectations for $H_4$, experienced and naïve birds led the

pair with approximately the same frequency in both exploration and exploitation, suggesting that deviations from established routes were not caused only by the naïve bird (see also *Appendix 1—figure 4*, inset). We did find evidence of behavioural asymmetries, in that transitions from exploitation to exploration were marginally more likely to be initiated by naïve birds (exact binomial test, p=0.042, *n* = 964, *Appendix 1—table 2*); however, this result was driven by those of generation 3 (p=0.02, *n* = 301), whereas no difference was detected in other generations. Transitions from exploration to exploitation were equally likely to be initiated by the two birds both overall and within each generation. However, when transitions are considered over the 12 releases composing each generation (*Figure 4c*), the naïve individual was more likely to initiate phases of exploration during the first release (p<0.01, *n* = 44, *Appendix 1—table 3*) doing so 70.5% of the time compared to 29.5% for the experienced bird. After the first release, transitions that initiate phases of exploration were about as likely to be initiated by either of the two birds independently of their level of experience.

## Discussion

For many group-living animals, searching for optimal travel routes can be a complex task as social information about routes can persist over generations regardless of its quality (*Helfman and Schultz, 1984*; *Sasaki and Biro, 2017*; *Jesmer et al., 2018*; *Laland and Williams, 1998*). This search is inherently subject to a trade-off between the exploitation of well-established route information accumulated over time and exploration for innovations that constitute potential improvements (*Hills et al., 2015*). Striking a balance is fundamental as a pronounced reliance on exploitation of learned information can hinder productive innovations (*Davies et al., 2012*) and thus promote the maintenance of potentially suboptimal behaviour and even of maladaptive behavioural traditions (*Laland and Williams, 1998*). Equally, an over-reliance on exploration without exploiting the rewards of beneficial innovations eventually impedes improvements in performance over time (*Fu and Gray, 2006*; *Mehlhorn et al., 2015*).

We studied the causal structure of this process in flights of the homing pigeon *C. livia* as this species is capable of both social learning and information pooling (*Biro et al., 2006*; *Pettit et al., 2013a*). Of particular interest for our study is the increase in route efficiency that results from the pairing of naïve individuals with experienced ones (*Pettit et al., 2013a*), including when this happens iteratively over multiple generations (*Sasaki and Biro, 2017*). Previous work has proposed information pooling as the underlying mechanism driving this increase in flight performance (*Sasaki and Biro, 2017*). Using transfer entropy to measure predictive information (*Schreiber, 2000*; *Pilkiewicz et al., 2020*), we found quantitative evidence that supports the information-pooling hypotheses $H_3$ and $H_4$ and in the strength of causal interactions within pairs of birds. Experienced and naïve birds influence each other's behaviour; about 20% of the future directional choices of any individual in a pair is explained by the behaviour of the other individual. These results contrast with our expectations for unidirectional social learning ($H_2$) that entails an asymmetric pattern of leadership with a pronounced role for experienced individuals.

Our analysis showed that, in a multi-generational transmission-chain design, the naïve bird has a higher influence than the experienced one during the early generations. In later generations, however, the experienced bird becomes the better source of predictive information. We hypothesize that, over generations, as birds have explored more search space and thus exhausted many alternative routes, a newly introduced bird becomes less likely to contribute productive innovations. As a result, innovations can lead to errors instead of improvements in later generations. Rather than leading to additional exploration bouts, these erroneous innovations could be suppressed by the contributions of the experienced individual that, as generations progress, has access to increasingly better information. This theoretical reasoning is analogous to the diminishing marginal value of returning to a previous location (i.e., continuing to rely on the naïve bird for innovations) when searching for an object in space (*Stone, 1976*). From an information-foraging perspective (*Stephens and Krebs, 1986*; *Pirolli, 2007*), the time invested in harvesting innovations on the introduction of a naïve bird corresponds to the time invested in attending to a newly discovered patch; at some point, the opportunity cost of further harvesting becomes too high to justify remaining in the patch. Thus, after the introduction of a naïve bird, the information-foraging pair shifts over successive releases from information-harvesting exploration back towards information-preserving exploitation, as would be expected in an optimal search problem (*Stone, 1976*). Over generations, as route information within the pair becomes better,

the balance between exploration for route improvement and exploitation of the known route changes in favour of the latter.

How do experimental pairs improve their homing routes over generations? Previous studies where leadership was defined on the basis of route similarity showed that, in pairs with a large difference in experience between birds, experienced individuals were more likely to assume leadership (*Flack et al., 2012*). Still using route similarity, *Sasaki and Biro, 2017* found evidence of social learning with naïve individuals learning routes from their experienced partners. Moreover, because newly formed pairs in the transmission-chain design also improved performance generation after generation, they proposed that naïve individuals could contribute innovations that pigeons evaluate in terms of route efficiency and prune away when inefficient (i.e., $H_3$). However, defining leadership in terms of causal interactions instead of route similarity allowed us to show that there is an asymmetric relation between innovators and exploiters only during the initial flight of a newly formed pair. Although leadership is ephemeral and equally shared between birds during exploration and exploitation independently of their level of experience, over the course of this first flight, the naïve individual disproportionally initiates phases of exploration, attracting the experienced bird to unfamiliar areas and triggering it to also resume the search. After that, both birds are equally likely to initiate transitions between exploration and exploitation as expected from our fourth hypothesis $H_4$. Moreover, as it is unlikely that experimental pairs were merely better than control groups at evaluating efficiency, we believe that their superior performance is rooted instead in their complex exploration–exploitation dynamics that allowed them to better cover the search space.

Personally acquired information allows solo individuals to improve flight performance (*Meade et al., 2005*) but only to discover routes moderately efficient (consistently within 0.8–0.85 efficiency across a large number of experiments, reviewed in *Guilford and Biro, 2014*) because solo individuals rapidly reduce their exploration efforts to seek out novel information. Experimental and fixed pairs of birds, on the other hand, explore more and thus outperform solo individuals. Together, two birds have superior sensory and cognitive capacities compared to single birds (*Krause et al., 2010*) which facilitates the discovery of better routes that are then learned collectively (*Biro et al., 2016*; *Kao et al., 2014*). The reasons why pairs explore more than solo individuals might lie partly with the conflicts characteristic of newly formed pairs (*Biro et al., 2006*) if the resolution of conflict, for example, through averaging individual inputs, indirectly prompts pairs to explore more and discover route innovations. This hypothesis could also explain why experimental pairs outperform fixed pairs. The introduction of a naïve individual at the start of each generation repeatedly creates an experience imbalance in the newly formed pair, which differs from fixed pairs that undergo a process of mutual habituation as they develop a stable route that likely reduces conflicts. This experience imbalance could be the source of new conflicts, providing a possible explanation for why experimental pairs reach levels of exploration generally higher than those of fixed pairs. The process of gradually settling on a joint route over the course of a generation following this initial perturbation is also reminiscent of the transient effects observed when an ant with outside information joins a group of nestmates transporting an object; the new information temporarily steers the collective in the right direction, but its effects on collective motion vanish quickly (*Gelblum et al., 2015*; *Gelblum et al., 2016*; *Gelblum et al., 2020*).

The ability of groups to outperform single individuals by pooling information across their members is an aspect of collective intelligence that has long intrigued researchers. One potential mechanism underlying this phenomenon, popularly known as the wisdom of crowds (*Surowiecki, 2005*), is averaging many individuals' estimates independent from each other. Averaging individual decisions is expected to provide a more accurate group estimate than any individuals' guess. Previous studies have also shown that animals can average their movement decisions to reach a compromise (*Biro et al., 2006*; *Strandburg-Peshkin et al., 2015*). Although the mechanisms by which experienced and naïve individuals pool information during route development remain unknown, our study points to the importance of naïve group members within the information-pooling process. Moreover, the wisdom of crowds is known to require personal information to be independent among group members (*Couzin, 2018*) otherwise, group performance can degrade quickly for increasing group size (*Kao and Couzin, 2014*). Experimental pairs could thus benefit from pooling information with naïve individuals that, at least at the beginning of each generation, likely provide a source of information independent from that of the experienced bird. The potentially deleterious effects of losing independence may provide another pressure to shift over time from innovative exploration to route-preserving exploitation. It

remains to be explored how our results generalize to larger flock sizes. Previous experiments without generational replacement showed that, even in larger flocks, birds flying ahead of the flock had a tendency to assume leadership positions (*Nagy et al., 2010*). However, the repeated introduction of naïve individuals into larger flocks might complicate the dichotomy between leaders and followers by inducing turnover dynamics between the front and the back of the flock.

Adopting an explicit exploration–exploitation perspective to study search strategies and doing so through the use of predictive information to quantify causal interactions has the potential to advance our understanding of collective navigation in larger flocks. The conceptual framework of exploration and exploitation as well as the methods we proposed can also benefit researchers studying other taxa that move in groups with the potential to learn from previous experiences, such as shoaling fish or certain primates. In principle, this information-theoretic approach could also be applied to the study of information transfer between the environment and the individuals within a group. The ability to quantify causal interactions of this sort could shed light on broader questions in ecology involving animals moving in a group and their environment such as the impact of visual landmarks on navigation or the effects of terrestrial migration on the environment (*Bracis and Mueller, 2017*; *de Guinea et al., 2021*).

## Materials and methods

Data and source code are available in *Valentini et al., 2021*.

### Experimental subjects and procedure

Data were taken from a previous study on cumulative route development in the homing pigeon *C. livia* (*Sasaki and Biro, 2017*). Pairs of birds composed of an *experienced* and a *naïve* bird were released together from the same site and allowed to fly back to the home loft. Pairs were created and released over five successive generations of a transmission chain, each generation lasting 12 consecutive releases of the same pair, according to the following procedure: initially, at generation 1, a naïve bird was released alone 12 times, allowing it to develop its own idiosyncratic route to the home loft. This bird, now experienced, was then paired with a new (naïve) bird at generation 2, and together they performed another 12 flights. This process was then repeated in the next generation with a new pair of birds composed of the former naïve bird and of a new naïve one. Data were gathered for a total of 10 independent transmission chains, each lasting five generations (see *Sasaki and Biro, 2017* for details). Birds flying at an average linear distance greater than 250 m from each other were not considered as pairs and were thus excluded from the analysis, leaving 343 flights with a mean ± SD flight duration of 8.65 ± 1.33 min. Additionally, in two control conditions, nine solo birds and six pairs (all initially naïve) were released from the same site for a total of 60 releases (the equivalent of 5 × 12 releases for the transmission chains).

### Data collection and pre-processing

Flight trajectories of birds were sampled at a frequency of 5 Hz using GPS loggers, converted from the geographic coordinate system to the metric system, and projected over the two-dimensional plane (see *Sasaki and Biro, 2017*). Each trajectory consisted of a time-ordered series of positions in space, $(x_i: (x^1, x^2)_i, i \geq 1)$, see *Figure 1* (point 1). We encoded the pattern of rotations of each flight using a binary symbolic representation where symbols 0 and 1 represent, respectively, a *clockwise* and a *counterclockwise* rotation (point 2). The direction of rotation was computed as the cross-product $\overrightarrow{x_{i-1}x_i} \times \overrightarrow{x_i x_{i+1}}$ between the motion vector at time $i$ and that at time $i + 1$. The rotation is clockwise when the product is negative and counterclockwise when it is positive. We also measured the distance $d_{EN}(i)$ of the experienced bird from the naïve one projected over the current direction of motion of the pair (cf. *Nagy et al., 2010* and Supplementary material). Using this distance, we then determined the relative position of birds over time: when $d_{EN} > 0$, the experienced bird was flying ahead of the naïve bird; when $d_{EN} < 0$, it was flying behind. Previous tests using the same GPS loggers showed that these devices have a sufficient level of accuracy with a small normally distributed spatial error (SD of 0.05 m) affecting the tracking accuracy of the direction of motion and a relatively larger error (median of 1.69 m) affecting that of the relative position (*Pettit et al., 2013b*). In our experiments, experienced

and naïve birds flew at an average distance from each other of 8.74 m with a standard deviation of 38.54 m.

## Measuring information transfer

We quantified the amount of information transferred between birds using information-theoretic measures (**Cover and Thomas, 2005**; **Pilkiewicz et al., 2020**) estimated from the series of rotations of the experienced, $E = (e_i, i \geq 1)$, and of the naïve, $N = (n_i, i \geq 1)$, birds (**Figure 1**, point 3). We aimed to quantify causal interactions between birds in a Wiener–Granger sense by measuring the extent to which the current behaviour of one bird allows us to predict the future behaviour of the other (**Bossomaier et al., 2016**). Here we describe the process for predicting the naïve bird's behaviour from that of the experienced one, but we also used the same method for the opposite direction. The average amount of information necessary to fully predict the next rotation of the naïve bird is quantified by the marginal entropy of its series of rotations $H(N^{i+1}) = -\sum n_i \, p(n_i) \, log_2 \, p(n_i)$ (**Figure 1**, Venn diagram, lower-left set). This is equal to bit if the flight of the naïve bird is maximally uncertain (i.e., clockwise and counterclockwise rotations are equally likely) and to bits if the flight is fully deterministic (i.e., rotations are either all clockwise or all counterclockwise). As a result of temporal autocorrelation (**Mitchell et al., 2019**), part of this information might be contained in the recent history of rotations of the naïve bird, $n_i^{(k)} = \{n_{i-k+1}, ..., n_{i-1}, n_i\}$ for history length $k$ (**Figure 1**, point 4, dark green entries in the naïve time series). The remaining predictive information, which is not explained by the past behaviour of the naïve bird, is quantified by the marginal entropy,

$$H(N^{i+1}|N^{(k)}) = - \sum_{n_i^{(k)}, n_{i+1}} p(n_i^{(k)}, n_{i+1}) \, \log_2 \frac{p(n_i^{(k)}, n_{i+1})}{p(n_i^{(k)})},$$

of its future rotations, $N^{i+1}$, conditioned on the outcome of the past rotations, (**Figure 1**, Venn diagram, overlap of the white and light purple areas).

Of interest to us was how much of this remaining information (necessary to predict the future direction of rotation of the naïve bird) can be obtained by the current behaviour of the experienced bird (**Figure 1**, points 4–6). This is given by the transfer entropy, which estimates the time-delayed effects on the naïve bird of its interaction with the experienced one: $T_{E \to N} = H(N^{i+1}|N^{(k)}) - H(N^{i+1}|N^{(k)}, E)$ (**Schreiber, 2000**). Transfer entropy is time directional, from the present of one bird to the future of the other, and considered for this reason a measure of information transfer (**Lizier and Prokopenko, 2010**). It is defined as

$$T_{E \to N} = \sum_{n_{i+1}, n_i^{(k)}, e_i} p(n_{i+1}, n_i^{(k)}, e_i) \, log_2 \frac{p(n_{i+1} \mid n_i^{(k)}, e_i)}{p(n_{i+1} \mid n_i^{(k)})}$$

and formally measures the reduction of uncertainty of the future rotation $N^{i+1}$ of the naïve bird given by knowledge of the current rotation of the experienced bird at the net of possible autocorrelations in the naïve bird's past $N^{(k)}$ (**Figure 1**, Venn diagram, area with red border). The logarithmic part of the above equation is known as local transfer entropy and measures over time whether the interaction at time (**Figure 1**, point 4, dark green and dark purple entries) was informative (positive) or misinformative (negative) (**Lizier et al., 2008**).

The above information-theoretic measures were computed in R 3.6.1 using the rinform-1.0.2 package (**Moore et al., 2018**) by estimating probabilities separately for each flight. To test whether causal interactions were significant, we also evaluated a surrogate dataset artificially created by pairing trajectories of birds that were not flown together: the trajectory of each experienced (or naïve) bird was paired with that of the naïve (or experienced) one from every other pair of birds not containing the same subjects.

## Measuring exploration and exploitation

For each point in a focal trajectory, we determined the point-to-point distance to a baseline trajectory as the minimum distance from the focal point to its closest point in the baseline trajectory. Using this measure, pairwise analysis of successive routes from the last three flights of the experienced bird during training (generation 1) showed that, once established, the bird largely remained within a point-to-point distance of 300 m from its idiosyncratic route (**Appendix 1—figure 3a**). Therefore, we used

300 m as a threshold to group points from a focal route into flight segments differentiated between those exploring new solutions and those exploiting known ones. We then compared the trajectories of consecutive flights for the same subjects to label each segment in the experimental and control datasets. For each focal trajectory that we aimed to label, we considered the trajectory of the previous release as a baseline trajectory for the comparison. In the case of paired birds (i.e., both experimental and fixed-pairs control), we considered the trajectory of the pair defined by the mean position of the two birds over time. For the first release of each generation in the experimental group, we used as baseline trajectory the last release of the previous generation because in this case there was no previous flight of the same pair to compare with. This approach to define exploration and exploitation led to a model of exploitation (i.e., the baseline trajectory) that varied over successive releases. Because the introduction of a naïve bird at each generation was likely to affect the baseline model of exploitation in a more pronounced manner than that of solo and fixed pairs of birds, this model might have been susceptible to differences between the experimental design of transmission chain experiments with respect to those of the two controls. To control for this scenario, we also explored an alternative approach where exploitation was defined on the basis of only information available to the experienced bird at the beginning of a new generation. In this case, the last release of the previous generation was used as the baseline trajectory for all releases within a given generation (see Supplementary methods). In both cases, we also measured the distance $d_{EN}(i)$ between experienced and naïve birds to determine which bird was flying at the front of the pair for a given route segment.

## Acknowledgements

GV, TPP, SIW, and SCP were supported by NSF grant no. PHY-1505048 awarded to SIW, TPP, and SCP. GV was also supported by research funds from Arizona State University to Prof. Bert Hölldobler. DB was supported by grant no. TWCF0316 from the Templeton World Charity Foundation's 'Diverse Intelligences' scheme. We thank Dr. Albert B Kao and Prof. Andrew Berdahl for helpful discussions.

## Additional information

### Funding

| Funder | Grant reference number | Author |
| --- | --- | --- |
| National Science Foundation | PHY-1505048 | Theodore P Pavlic<br>Sara Imari Walker<br>Stephen C Pratt |
| Templeton World Charity Foundation | TWCF0316 | Dora Biro |

The funders had no role in study design, data collection and interpretation, or the decision to submit the work for publication.

### Author contributions

Gabriele Valentini, Conceptualization, Formal analysis, Investigation, Methodology, Software, Visualization, Writing - original draft, Writing – review and editing; Theodore P Pavlic, Sara Imari Walker, Stephen C Pratt, Funding acquisition, Writing – review and editing; Dora Biro, Funding acquisition, Investigation, Resources, Writing – review and editing; Takao Sasaki, Conceptualization, Investigation, Resources, Writing – review and editing

### Author ORCIDs

Gabriele Valentini (ID) http://orcid.org/0000-0002-8961-3211
Theodore P Pavlic (ID) http://orcid.org/0000-0002-7073-6932
Sara Imari Walker (ID) http://orcid.org/0000-0001-5779-2772
Stephen C Pratt (ID) http://orcid.org/0000-0002-1086-4019

### Decision letter and Author response

Decision letter https://doi.org/10.7554/eLife.68653.sa1
Author response https://doi.org/10.7554/eLife.68653.sa2

## Additional files

### Supplementary files
• Transparent reporting form

### Data availability
Data and code have been deposited in figshare with identifier "10.6084/m9.figshare.14043362".

The following dataset was generated:

| Author(s) | Year | Dataset title | Dataset URL | Database and Identifier |
|---|---|---|---|---|
| Valentini G, Pavlic T, Walker SI, Pratt SC, Biro D, Sasaki T | 2021 | Data and code from: Naïve individuals promote collective exploration in homing pigeons | https://figshare.com/articles/dataset/Data_and_code_from_Na_ve_individuals_promote_collective_exploration_in_homing_pigeons/14043362 | figshare, 10.6084/m9.figshare.14043362.v1 |

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

## Appendix 1

### Methods

### Relative position of birds over time

We determined the relative position of each bird within the pair using the distance $d_{EF}$ of the experienced bird from the naïve one projected onto the direction of motion of the flock. We modified the method proposed in **Nagy et al., 2010**, which gives the distance of the experienced bird from the centre of the flock, to also include the segment from the centre of the flock to the naïve bird. This is given by

$$d_{EN}(i) = (\vec{x_E}(i) - \vec{x_E}(i)) \cdot \vec{v_{pair}}(i) \cdot 2,$$

where $\vec{x_E}(i)$ and $\vec{x_N}(i)$ are the positions of the experienced and the naïve bird and $\vec{v_{pair}}(i)$ is the normalized velocity of the pair. The normalized velocity is computed as

$$\vec{v_{pair}}(i) = \frac{\left\langle \vec{\dot{x}_k}(i) \right\rangle_k}{\left| \left\langle \vec{\dot{x}_k}(i) \right\rangle_k \right|}$$

As the flock is composed of two birds only, the projected distance of the experienced bird from the naïve one projected onto the direction of motion of the pair is positive, when the experienced bird is flying *ahead* of the naïve one, and it is negative when the experienced bird is flying *behind*.

### Exploration–exploitation with respect to the previous generation

Our primary approach to define exploration and exploitation is based on comparing pairs of successive releases using the trajectory of the previous release as our baseline model of exploitation and then labelling newer portions of a focal route as exploration. The model of exploitation (i.e., baseline trajectory) thus varies for each focal release similarly to a moving-average window over successive releases. An alternative approach to define exploration and exploitation is to consider a constant model of exploitation for each release within a given generation. This can be obtained by setting the baseline trajectory to equal the last trajectory of the previous generation. As a consequence, exploitation is defined on the basis of the information available only to the experienced bird at the beginning of a new generation; every portion of the route within that generation that is more than 300 m away from the baseline is considered exploration.

At the first generation of the experimental group, when birds are trained individually for the successive transmission chain experiment and there is no previous generation available to provide us with a baseline trajectory, we consider the first release as the baseline trajectory for the remainder of the generation. In the case of solo and fixed-pairs controls, for which there is no obvious definition of a generation, we considered the 60 releases in each of these experiments as formed by five generations, each lasting 12 releases, and used an equivalent approach to that of the experimental group to define exploration and exploitation.

### Results

### Landscape of information transfer and choice of parameters

We first explored how information transfer between the birds varied as a function of two parameters: the sampling period (seconds) and the history length used in the computation of transfer entropy. Our original GPS data are sampled at a frequency of one sample every 0.2 s (5 Hz); we can further subsample these data by dropping samples, for example, using one sample every 0.4, 0.6,..., 4.0 s as the sampling period. The history length represents the number of past rotations by the bird we want to predict that we consider when computing transfer entropy from the other bird in the pair.

The total transfer of information between the birds varies as a function of these parameters (**Appendix 1—figure 1a**). It peaks in the region delimited by history lengths of 8–10 time steps and a sampling period between 0.2 and 1.2 s, whereas it vanishes otherwise. The total transfer of information, which is generally adopted as a measure to choose study parameters (**Porfiri, 2018**), reaches its maximum for a history length of $k = 10$ at one sample every 0.2 s (black triangle in **Appendix 1—figure 1a**). We use this parameter configuration for the rest of our information-theoretic analysis. For this and for most other parameter configurations, the naïve bird is more informative about the future behaviour of the experienced one than the other way around

(*Appendix 1—figure 1b*). Only in two regions, both far from the point maximizing the total transfer of information, is the experienced bird more informative than the naïve one; however, the overall amount of information transferred between birds in these regions is negligible.

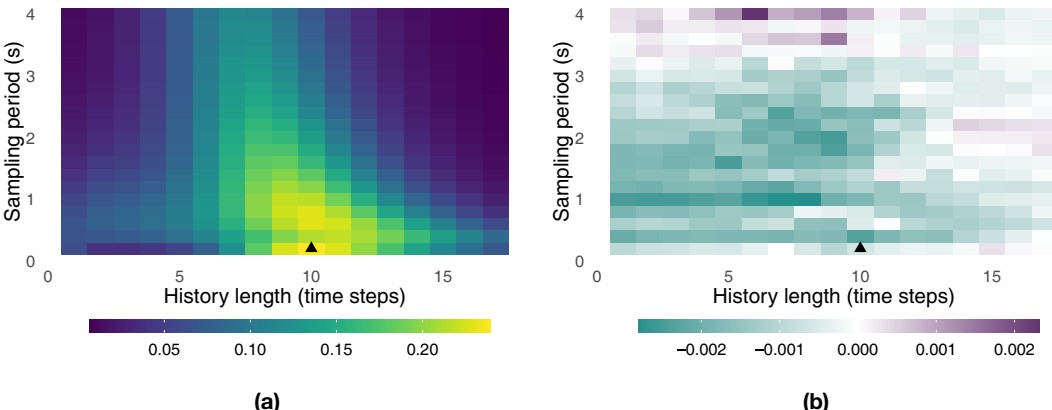

**Appendix 1—figure 1.** Landscape of information transfer as a function of the history length, $k = \in \{1, \dots, 17\}$, and of the sampling period, $\{0.2, 0.4, \dots, 4.0\}$ s. Panel (**a**) shows the total transfer of information between the pair of birds, $T_{E \to N} + T_{N \to E}$, averaged over all releases and generations. Panel (**b**) shows the net transfer of information between the pair of birds, $T_{E \to N} - T_{N \to E}$, averaged over all releases and generations. Positive (respectively, negative) values represent configurations where the experienced (naïve) bird is more informative than the naïve (experienced) one. The triangle represents the configuration with maximum total transfer of information.

## Comparison of information transfer with the surrogate dataset

To verify their significance, we compared our estimates of information transfer between the two birds with equivalent measurements taken from the surrogate dataset. We expect causal effects measured in the original dataset to be stronger than those found in the surrogate one. As shown in *Appendix 1—table 1*, our expectations are fully met: the original dataset shows values of transfer entropy significantly higher than those observed in the surrogate dataset, both for the entire dataset as well as for each separate generation.

**Appendix 1—table 1.** Statistical comparison of information transfer between the original and the surrogate dataset over all generations and over separate generations.

Column 1 reports the generation and sample sizes. Columns 2 and 4 report the differences between the mean value of transfer entropy of the original dataset and that of the surrogate dataset. Columns 3 and 5 report the results of one-sided two-sample Whitney–Mann–Wilcoxon rank-sum tests with continuity correction (p-value and $W$ statistic) testing if the original dataset has significantly higher transfer entropy than the surrogate one. Significant p-values are reported in bold.

**Original vs. surrogate dataset**

| Generation | $T_{E \to N} - T_{E \to N}^s$ | $H_1 : T_{E \to N} > T_{E \to N}^s$ | $T_{N \to E} - T_{N \to E}^s$ | $H_1 : T_{N \to E} > T_{N \to E}^s$ |
|---|---|---|---|---|
| All $(n = 343, n^s = 29035)$ | $\mu = 0.0089$ | $\boldsymbol{p < .001}\ (W = 6145522)$ | $\mu = 0.0088$ | $\boldsymbol{p < .001}\ (W = 6126284)$ |
| 2 $(n = 94, n^s = 7912)$ | $\mu = 0.0062$ | $\boldsymbol{p < .001}\ (W = 445262)$ | $\mu = 0.0074$ | $\boldsymbol{p < .001}\ (W = 452733)$ |
| 3 $(n = 99, n^s = 9801)$ | $\mu = 0.0094$ | $\boldsymbol{p < .001}\ (W = 615721.5)$ | $\mu = 0.0119$ | $\boldsymbol{p < .001}\ (W = 645665.5)$ |
| 4 $(n = 81, n^s = 6561)$ | $\mu = 0.0071$ | $\boldsymbol{p = .006}\ (W = 308433.5)$ | $\mu = 0.0057$ | $\boldsymbol{p = .015}\ (W = 303085.5)$ |
| 5 $(n = 69, n^s = 4761)$ | $\mu = 0.0111$ | $\boldsymbol{p < .001}\ (W = 214258.5)$ | $\mu = 0.0075$ | $\boldsymbol{p = .002}\ (W = 197618.5)$ |

## Analysis of time spent by each bird at the front of the pair

We divided each flight into different segments, with each segment representing a consecutive portion of the route with either the experienced or the naïve bird at the front of the pair, and then measured the segment durations (**Appendix 1—figure 2**). The distribution of segment durations resembles a log-normal distribution for both the entire dataset and for individual generations. Experienced and naïve birds are characterized by very similar distributions closely overlapping each other. Overall, the naïve bird spent significantly longer periods of time at the front of the pair (**Appendix 1—table 2**), but the difference is small and largely driven by that of generation 3 (the only generation showing significant differences). With the exception of generation 3, the experienced and the naïve bird spent, on average, an approximately equal portion of the route at the front of the pair (**Appendix 1—table 3**). At generation 3, the naïve bird was at the front of the pair for a significantly larger portion of the route with respect to the experienced bird (respectively, 54% vs. 46%).

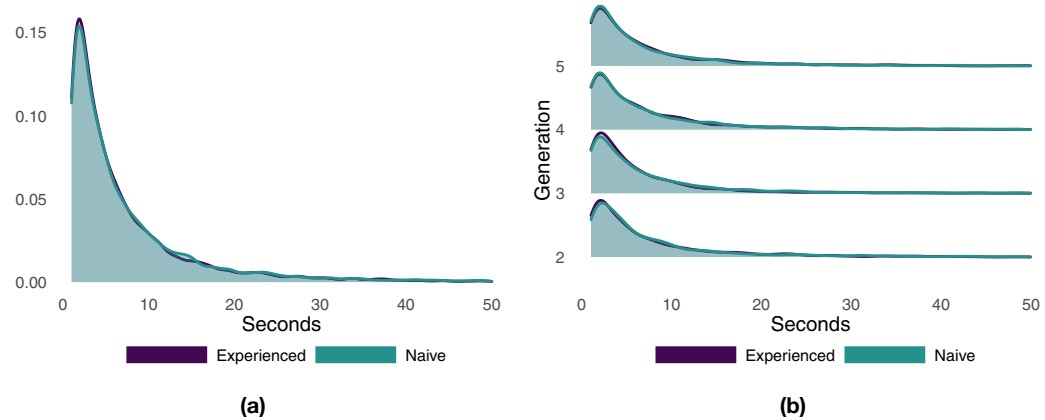

(a) (b)

**Appendix 1—figure 2.** Probability density function of the duration of flight segments with either the experienced or the naïve bird at the front of the pair. Panel (**a**) shows the results aggregated over all generations. Panel (**b**) shows the results separately for each generation.

**Appendix 1—table 2.** Statistics for the duration of flight segments with either the experienced or the naïve bird at the front of the pair.

Column 1 reports the generation and sample sizes. Columns 2 and 3 give the mean duration and the standard deviation of segments for the experienced and the naïve bird. Column 4 reports the results of two-sided two-sample Whitney–Mann–Wilcoxon rank-sum tests with continuity correction (p-value and $W$ statistic) testing differences between the distribution $D$ of the duration of flight segments for the two birds. Significant p-values are reported in bold.

| Generation | Experienced | Naïve | $H_1 : D_E \neq D_N$ |
|---|---|---|---|
| All $\left(n^E = 10725, n^N = 10773\right)$ | $\mu = 7.93, \sigma = 12.92$ | $\mu = 8.23, \sigma = 14.12$ | $\boldsymbol{p = .047}\ (W = 56865638)$ |
| $2\left(n^E = 2512, n^N = 2492\right)$ | $\mu = 8.89, \sigma = 17.51$ | $\mu = 8.81, \sigma = 13.55$ | $p = .13\ (W = 3052262)$ |
| $3\left(n^E = 3690, n^N = 3737\right)$ | $\mu = 6.83, \sigma = 9.08$ | $\mu = 7.96, \sigma = 14.51$ | $\boldsymbol{p < .001}\ (W = 6588674)$ |
| $4\left(n^E = 2297 n^N = 2283\right)$ | $\mu = 8.63, \sigma = 13.65$ | $\mu = 8.55, \sigma = 14.23$ | $p = .86\ (W = 2630132)$ |
| $5\left(n^E = 2226, n^N = 2261\right)$ | $\mu = 7.98, \sigma = 11.23$ | $\mu = 7.73, \sigma = 13.93$ | $p = .23\ (W = 2568854)$ |

**Appendix 1—table 3.** Statistics for the proportion of a flight with either the experienced or the naïve bird at the front of the pair.

Column 1 reports the generation and sample size. Columns 2 and 3 give the mean and the standard deviation of the proportion of a flight with either the experienced or the naïve bird at the front of the pair. Column 4 reports the results of two-sided paired Wilcoxon signed-rank tests with continuity

correction (p-value and *V* statistic) testing differences between the distribution of the proportion *P* of a flight for the two birds. Significant p-values are reported in bold.

| Generation | Experienced | Naïve | $H_1 : P_E \neq P_N$ |
|---|---|---|---|
| All $(n = 341)$ | $\mu = 0.49, \sigma = 0.2$ | $\mu = 0.51, \sigma = 0.2$ | $p = .46\ (V = 27817.5)$ |
| 2 $(n = 92)$ | $\mu = 0.51, \sigma = 0.23$ | $\mu = 0.49, \sigma = 0.23$ | $p = .69\ (V = 2243)$ |
| 3 $(n = 99)$ | $\mu = 0.46, \sigma = 0.17$ | $\mu = 0.54, \sigma = 0.17$ | $\boldsymbol{p = .03}\ (V = 1851)$ |
| 4 $(n = 81)$ | $\mu = 0.5, \sigma = 0.2$ | $\mu = 0.5, \sigma = 0.2$ | $p = .84\ (V = 1705)$ |
| 5 $(n = 69)$ | $\mu = 0.49, \sigma = 0.2$ | $\mu = 0.51, \sigma = 0.2$ | $p = .95\ (V = 1219)$ |

## Exploration versus exploitation with respect to successive releases

We used a distance-based mechanism to label portions of a focal route as either exploration or exploitation depending on the point-to-point distance from each point of the focal route to the closest point of the baseline route at the previous release. To determine a suitable threshold, we compared successive trajectories flown by the experienced bird towards the end of training (generation 1, last three flights) and studied the distribution of distances between successive trajectories (*Appendix 1—figure 3a*). The distribution of distances is right-skewed and approximately exponential. A threshold of 300 m captures a large portion of the probability mass, about 70.5%, whereas larger distances are progressively less likely. On this basis, we set a threshold of 300 m to distinguish between phases of exploitation (<300 m) and phases of exploration (≥300 m). *Appendix 1—figure 3b* shows the distribution of distances between consecutive flights observed during each generation, whereas *Appendix 1—figure 4a* shows the same results aggregated over all generations with details of the proportion of time that each bird is leading as a function of the same distance showed in *Appendix 1—figure 4d*. *Appendix 1—figure 4b and c* show similar results for fixed pairs of birds and solo individuals. Whereas fixed pairs are characterized by a distribution similar to that of experimental pairs, solo birds have a markedly shifted distribution towards exploitation at the expense of exploration.

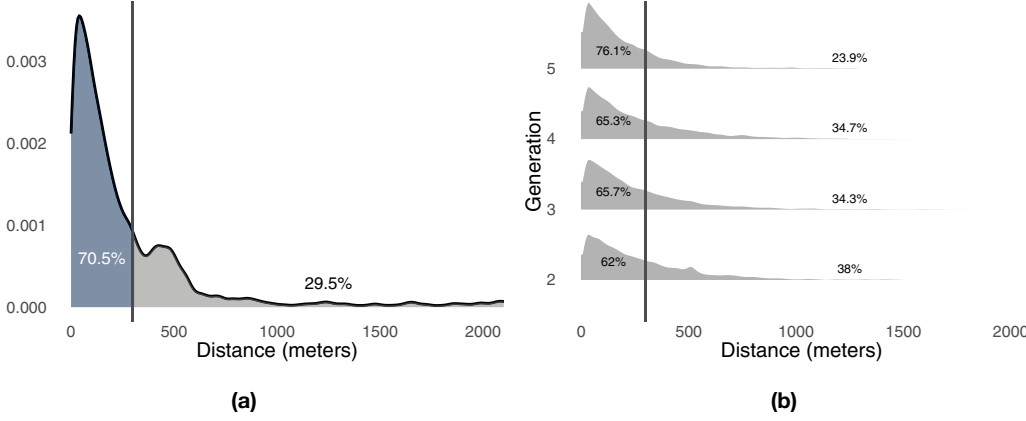

**Appendix 1—figure 3.** Distribution of minimum distances between the pairs of consecutive flights. Panel (**a**) shows the results for the trained birds during the first generation of the experiment. Panel (**b**) shows the results for the pair of birds during all remaining generations of the experiments. Vertical lines highlight the division of the probability mass between exploitation (left) and exploration (right) defined by a 300 m threshold.

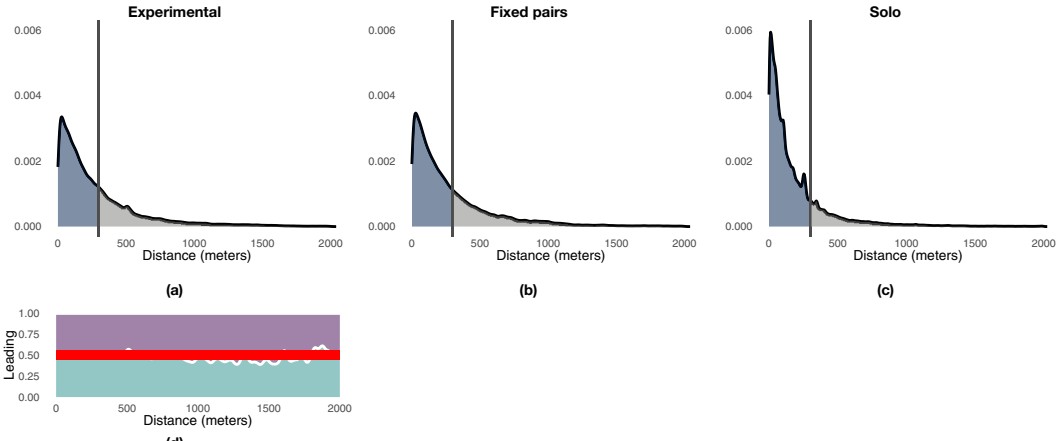

**Appendix 1—figure 4.** Illustration of the distribution of point-to-point distances between pairs of consecutive flights highlighting the 300 m threshold that demarks the end of exploitation and the beginning on exploration. Panel (**a**) reports the results for experimental pairs over generations 2–5, panel (**b**) reports those for the fixed-pairs control, and panel (**c**) those for the solo control. Panel (**d**) shows the proportion of times experienced (purple) and naïve (green) birds are leading the flock as a function of the distance between the current and the previous trajectory. The red line indicates an equal likelihood between the two birds to lead the flock.

We compared the proportion of exploration and exploitation across experimental conditions before the beginning (i.e., generation 1) and during transmission chain experiments (i.e., generations 2–5). During the first 12 releases (*Appendix 1—table 4*), individuals from the experimental group that flew solo during generation 1 could not be distinguished from birds in the solo control, whereas fixed pairs of birds showed levels of exploration significantly higher than those of solo birds. During releases 13–60, that is, when naïve individuals are iteratively introduced in the transmission chains at the beginning of each generation, experimental pairs showed instead significantly higher levels of exploration than both solo and fixed pairs of birds, with these former exploring much less than birds that flew in pairs.

**Appendix 1—table 4.** Statistical comparison of mean proportions of a flight spent exploring versus exploiting across treatments (experimental pairs, solo, and fixed-pairs controls) for the first 12 releases (generation 1) and for releases 13–60 (generation 2–5).
Entries report the proportion of exploration vs. exploitation for pairs of treatments as well as the results of two-sided two-sample Whitney–Mann–Wilcoxon rank-sum tests with continuity correction (p-value and *W* statistic) for differences in proportion of exploration. Significant p-values are reported in bold. Results of testing for the proportion of exploitation are equivalent and not repeated below.

| Releases | Dataset | Solo control | Fixed-pairs control |
|---|---|---|---|
| | | | Row: 36.7% vs. 63.3%<br>Col: 51.7% vs. 48.3%<br>**p <. 001** (W = 2230) |
| | | Row: 36.7% vs. 63.3%<br>Col: 34.2% vs. 65.8% | |
| | Experimental (generation 1) | | |
| | | | Row: 36.7% vs. 63.3%<br>Col: 34.2% vs. 65.8%<br>**p <.001** (W=1837) |
| 1–12 | Solo control | – | |

*Appendix 1—table 4 Continued on next page*

*Appendix 1—table 4 Continued*

| Releases | Dataset | Solo control | Fixed-pairs control |
|---|---|---|---|
| | Experimental (generations 2–5) | Row: 32.9% vs. 67.1%<br>Col: 15.7% vs. 84.3% | Row: 32.9% vs. 67.1%<br>Col: 29.3% vs. 70.7%<br>**p=.0456** (W=50472) |
| | | | Row: 15.7% vs. 84.3%<br>Col: 29.3% vs. 70.7%<br>**p<.001** (W=31517) |
| 13–60 | Solo control | – | |

## Exploration versus exploitation with respect to the previous generation

In addition to the exploration–exploitation analysis with respect to successive releases, we also performed a similar analysis where exploitation is defined with respect to the previous generation (see supplementary Methods above) in order to validate the robustness of the main results obtained with our primary approach. The distributions of point-to-point distances between baseline and focal trajectories (*Appendix 1—figure 5*) resembled those observed when comparing successive releases (*Appendix 1—figure 4*). Differences across experimental and control treatments (*Appendix 1—table 5*) were much more pronounced under this model but in line with the results obtained above (*Appendix 1—table 4*). Moreover, the overall exploration trends reported in *Appendix 1—figure 6* showed the same signatures of the analysis over successive releases (*Figure 4a*). Exploration decreased over generations in all experimental conditions with experimental pairs exploring more than both fixed pairs of birds and solo individuals.

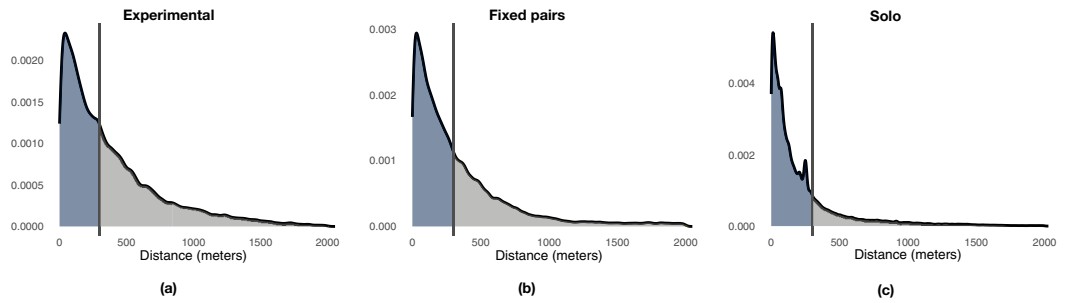

**Appendix 1—figure 5.** Illustration of the distribution of point-to-point distances between each flight of a generation (focal trajectories) and the last flight of the previous generation (baseline trajectory). Colors and vertical lines highlight the 300 meter threshold that demarks the end of exploitation and the beginning of exploration. Panel (**a**) reports the results for experimental pairs over generations 2–5, panel (**b**) reports those for the fixed pairs control, and panel (**c**) those for the solo control.

**Appendix 1—table 5.** Statistical comparison of mean proportions of a flight spent exploring versus exploiting across treatments (experimental pairs, solo and fixed pairs controls) for the first 12 releases (generation 1) and for releases 13–60 (generation 2–5) when considering the last release at the previous generation as the baseline trajectory.
Entries report the proportion of exploration vs exploitation for pairs of treatments as well as the results of two-sided two-sample Whitney–Mann–Wilcoxon rank-sum tests with continuity correction (-value and statistic) for differences in proportion of exploration. Significant -values are reported in bold. Results of testing for the proportion of exploitation are equivalent and not repeated below.

| Releases | Dataset | Solo control | Fixed pairs control |
|---|---|---|---|
| 1–12 | Experimental (gen. 1) | Row: 61.5 % vs 38.5%<br>Col: 55.8 % vs 44.2% | Row: 61.5 % vs 38.5%<br>Col: 82.9 % vs 17.1% |
| | Solo control | – | Row: 55.8 % vs 44.2%<br>Col: 82.9 % vs 17.1% |

*Appendix 1—table 5 Continued on next page*

*Appendix 1—table 5 Continued*

| Releases | Dataset | Solo control | Fixed pairs control |
|---|---|---|---|
| 13–60 | Experimental (gen. 2–5) | Row: 46.6 % vs 53.4%<br>Col: 18.9 % vs 81.1% | Row: 46.6 % vs 53.4%<br>Col: 32.4 % vs 67.6% |
| | Solo control | – | Row: 18.9 % vs 81.1%<br>Col: 32.4 % vs 67.6% |

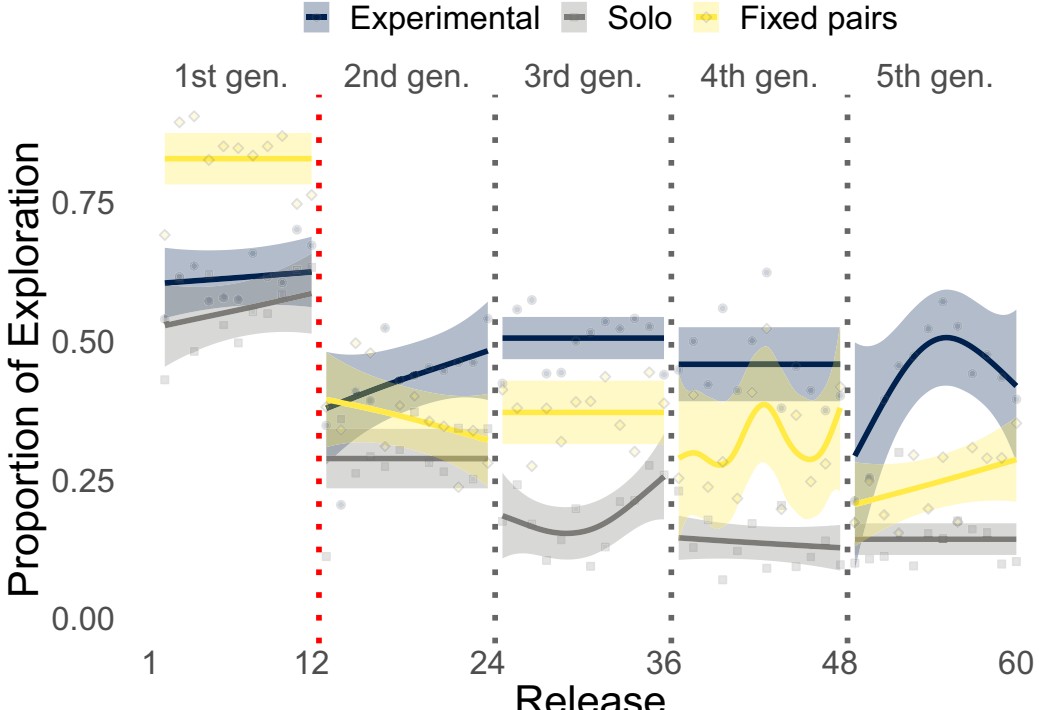

**Appendix 1—figure 6.** Illustration of the proportion of exploration (respectively, one minus the proportion of exploitation) over releases when considering the last release at the previous generation as the baseline trajectory. Results are shown for the experimental group, the solo control, and the fixed pairs control. The red dotted vertical line delineates the end of the experimental group's training phase. Smoothed lines are computed with generalized additive models using shrinkage cubic regression splines (mean and standard error); points represent averages for individual releases.

## The role of leadership during exploration and exploitation

Using this distance-based mechanism, we label each segment of a flight as either exploration or exploitation. For each segment, we keep track of which bird, experienced or naïve, is leading the pair. We then compute the proportion of time spent by the experienced bird and by the naïve one leading the pair during either an exploration or exploitation phase (*Appendix 1—table 6*). The combined proportion of time of both birds in either exploration or exploitation gives instead the corresponding contribution of the pair. Overall, each bird spends approximately 16.5 % of the time leading the pair during an exploration phase and 33.5 % of the time leading the pair during an exploitation phase. With the exception of exploitation phases in generation 3, there is no significant difference between the proportion of time spent by the two bird in an exploration or exploitation phase (*Appendix 1—table 7*).

**Appendix 1—table 6.** Proportion of a flight led by each of the two birds, calculated separately for exploration and exploitation phases.

Column one reports the generation and sample size. Columns 2 and 3 give the mean and the standard deviation of the proportion of a flight led, respectively, by the experienced and the naïve bird for the case of exploration. Columns 4 and 5 give the mean and the standard deviation of the proportion of a flight led, respectively, by the experienced and the naïve bird for the case of exploitation.

| | Exploration | | Exploitation | |
|---|---|---|---|---|
| Generation | Experienced | Naïve | Experienced | Naïve |
| All $(n = 341)$ | $\mu = 0.17, \sigma = 0.15$ | $\mu = 0.16, \sigma = 0.13$ | $\mu = 0.32, \sigma = 0.18$ | $\mu = 0.35, \sigma = 0.2$ |
| 2 $(n = 92)$ | $\mu = 0.2, \sigma = 0.17$ | $\mu = 0.17, \sigma = 0.14$ | $\mu = 0.31, \sigma = 0.2$ | $\mu = 0.32, \sigma = 0.21$ |
| 3 $(n = 99)$ | $\mu = 0.16, \sigma = 0.13$ | $\mu = 0.19, \sigma = 0.13$ | $\mu = 0.3, \sigma = 0.16$ | $\mu = 0.36, \sigma = 0.17$ |
| 4 $(n = 81)$ | $\mu = 0.18, \sigma = 0.16$ | $\mu = 0.16, \sigma = 0.13$ | $\mu = 0.32, \sigma = 0.18$ | $\mu = 0.34, \sigma = 0.22$ |
| $(n = 69)$ | $\mu = 0.13, \sigma = 0.13$ | $\mu = 0.1, \sigma = 0.1$ | $\mu = 0.37, \sigma = 0.17$ | $\mu = 0.4, \sigma = 0.21$ |

**Appendix 1—table 7.** Statistical comparison of leadership by experienced vs. naïve birds, tested separately for exploration and exploitation over all generations and over separate generations. Column one reports the generation and sample size. Columns 2 and 3 report the results of two-sided paired Wilcoxon signed-rank tests with continuity correction (-value and statistic) for differences in proportion of flight led between experienced and naïve birds for exploration and exploitation, respectively. Significant -values are reported in bold.

| | Experienced vs naïve | |
|---|---|---|
| Generation | Exploration | Exploitation |
| All $(n = 341)$ | $p = .44 \ (V = 28820)$ | $p = .13 \ (V = 26240)$ |
| 2 $(n = 92)$ | $p = .17 \ (V = 2490)$ | $p = .71 \ (V = 2048)$ |
| 3 $(n = 99)$ | $p = .08 \ (V = 1891)$ | $\boldsymbol{p = .035} \ (V = 1871)$ |
| 4 $(n = 81)$ | $p = .36 \ (V = 1767)$ | $p = .79 \ (V = 1603)$ |
| 5 $(n = 69)$ | $p = .22 \ (V = 1187)$ | $p = .8 \ (V = 1131)$ |

## Transitions between phases of exploration and exploitation

Although once initiated, phases of exploration and phases of exploitation are led in equal manner by the experienced and the naïve bird, the experience unbalance within the pair might affect the likelihood of a bird to initiate transitions from one phase of the exploration–exploitation process to other. To investigate this question, we measured transition probabilities for both birds over generations (*Appendix 1—table 8*) and over releases (*Appendix 1—table 9*). Transitions from exploration to exploitation are not significantly different between experienced and naïve birds, but transitions from exploitation to exploration are significantly (albeit marginally) more likely to be initiated by naïve birds. However, this result seems driven by the data of generation 3, where the naïve individual is marginally but significantly more likely than the experienced one to initiate exploration phases, while in all other generations both birds are equally likely to initiate changes in either direction. When looking at transition probabilities over releases, we found a similar trend except for the first release where the naïve bird is much more likely to initiate exploration phases.

**Appendix 1—table 8.** Statistical comparison of the proportion of transitions from exploitation to exploration and from exploration to exploitation led by the experienced and by the naïve bird over generations.
Column one reports the generation number. Columns 2 and 4 report the estimated probabilities that transitions are led by the experienced bird,, and by the naïve bird,, for transitions, respectively, from exploitation to exploration and from exploration to exploitation. Columns 3 and 5 give the results of exact binomial tests (-value and sample size) of the null hypothesis that the probability that transitions are led by the experienced bird equals 0.5. Significant -values are reported in bold.

| | Exploitation $\rightarrow$ Exploration | | Exploration $\rightarrow$ Exploitation | |
|---|---|---|---|---|
| Generation | $P_E \, versus \, P_N$ | $H_1 : P_E \neq 0.5$ | $P_E \, versus \, P_N$ | $H_1 : P_E \neq 0.5$ |

*Appendix 1—table 8 Continued*

| | **Exploitation → Exploration** | | **Exploration → Exploitation** | |
|---|---|---|---|---|
| All | $P_E = 0.467, P_N = 0.533$ | **p = .042** $(n = 964)$ | $P_E = 0.513, P_N = 0.487$ | $p = .42$ $(n = 966)$ |
| 2 | $P_E = 0.494, P_N = 0.506$ | $p = .9$ $(n = 247)$ | $P_E = 0.52, P_N = 0.48$ | $p = .56$ $(n = 244)$ |
| 3 | $P_E = 0.432, P_N = 0.568$ | **p = .02** $(n = 301)$ | $P_E = 0.483, P_N = 0.517$ | $p = .6$ $(n = 300)$ |
| 4 | $P_E = 0.5, P_N = 0.5$ | $p = 1$ $(n = 216)$ | $P_E = 0.539, P_N = 0.461$ | $p = .28$ $(n = 219)$ |
| 5 | $P_E = 0.45, P_N = 0.55$ | $p = .18$ $(n = 200)$ | $P_E = 0.522, P_N = 0.478$ | $p = .57$ $(n = 203)$ |

**Appendix 1—table 9.** Statistical comparison of the proportion of transitions from exploitation to exploration and from exploration to exploitation led by the experienced and by the naïve bird over releases.

Column one reports the release number. Columns 2 and 4 report the estimated probabilities that transitions are led by the experienced bird,, and by the naïve bird,, for transitions, respectively, from exploitation to exploration and from exploration to exploitation. Columns 3 and 5 give the results of exact binomial tests (-value and sample size) of the null hypothesis that the probability that transitions are led by the experienced bird equals 0.5. Significant -values are reported in bold.

| | **Exploitation → Exploration** | | **Exploration → Exploitation** | |
|---|---|---|---|---|
| *Release* | $P_E versus P_N$ | $H_1 : P_E \neq 0.5$ | $P_E versus P_N$ | $H_1 : P_E \neq 0.5$ |
| 1 | $P_E = 0.295, P_N = 0.705$ | **p < .01** $(n = 44)$ | $P_E = 0.477, P_N = 0.523$ | $p = .88$ $(n = 44)$ |
| 2 | $P_E = 0.478, P_N = 0.522$ | $p = .81$ $(n = 69)$ | $P_E = 0.522, P_N = 0.478$ | $p = .81$ $(n = 67)$ |
| 3 | $P_E = 0.519, P_N = 0.481$ | $p = .83$ $(n = 81)$ | $P_E = 0.575, P_N = 0.425$ | $p = .22$ $(n = 80)$ |
| 4 | $P_E = 0.576, P_N = 0.424$ | $p = .27$ $(n = 66)$ | $P_E = 0.6, P_N = 0.4$ | $p = .14$ $(n = 65)$ |
| 5 | $P_E = 0.487, P_N = 0.513$ | $p = .91$ $(n = 76)$ | $P_E = 0.519, P_N = 0.481$ | $p = .82$ $(n = 77)$ |
| 6 | $P_E = 0.54, P_N = 0.46$ | $p = .48$ $(n = 100)$ | $P_E = 0.465, P_N = 0.535$ | $p = .55$ $(n = 99)$ |
| 7 | $P_E = 0.443, P_N = 0.557$ | $p = .34$ $(n = 88)$ | $P_E = 0.494, P_N = 0.506$ | $p = 1.0$ $(n = 89)$ |
| 8 | $P_E = 0.409, P_N = 0.591$ | $p = .18$ $(n = 66)$ | $P_E = 0.597, P_N = 0.403$ | $p = .14$ $(n = 67)$ |
| 9 | $P_E = 0.438, P_N = 0.562$ | $p = .22$ $(n = 112)$ | $P_E = 0.496, P_N = 0.504$ | $p = 1$ $(n = 115)$ |
| 10 | $P_E = 0.422, P_N = 0.578$ | $p = .17$ $(n = 90)$ | $P_E = 0.522, P_N = 0.478$ | $p = .75$ $(n = 92)$ |
| 11 | $P_E = 0.391, P_N = 0.609$ | $p = .053$ $(n = 87)$ | $P_E = 0.453, P_N = 0.547$ | $p = .45$ $(n = 86)$ |
| 12 | $P_E = 0.541, P_N = 0.459$ | $p = .52$ $(n = 85)$ | $P_E = 0.482, P_N = 0.518$ | $p = .83$ $(n = 85)$ |

