## [Editor Report]

This study in the field of collective behavior addresses how naïve and experienced individuals (i.e., homing pigeons) pool information in order to navigate while flying back home. The authors show that the passage of information is largely democratic, meaning information passes both ways, and that, unexpectedly, exploration of the route is initiated both by naïve and experienced birds. The work provides a new perspective on information sharing during collective learning.

---

## [Decision Letter]

**Decision letter after peer review:**

Thank you for submitting your article "Naïve individuals promote collective exploration in homing pigeons" for consideration by *eLife*. Your article has been reviewed by 2 peer reviewers, and the evaluation has been overseen by a Reviewing Editor and Detlef Weigel as the Senior Editor. The reviewers have opted to remain anonymous.

Essential revisions:

1) Better explain how the point-to-point distance for exploration/exploitation was computed (Referee 1).

2) Discuss why the experienced birds explores more as they move further in generations (Referee 1).

3) Extend the introductory paragraph on Transfer Entropy.

4) Discuss connections with research on leadership and information pooling in other animal groups and add citations to the relevant bibliography.

*Reviewer #1 (Recommendations for the authors):*

The paper is fairly straightforward to read and beyond a few general comments and methodological questions I have no problem in this being accepted. The authors have done a great job in analyzing a rich dataset to address an important question in the field.

Methodology:

I did not quite understand how the point-to-point distance for exploration/exploitation was measured (lines 231-233, and lines 110-112 in Supplementary). Were these distances computed as maximum distance between two trajectories within a time window, were they max between trajectories at all time steps? if yes, what time steps. I think a figure in the Supplementary would help

Discussion:

– Can the authors say something about what do these results imply for larger flocks? What would be the limitations of this approach (would the leader bird still be in front for example?)

– What do the authors think is special about generation 3 that it does not follow the trend in several results?

– Why do the authors think that the experienced bird starts exploring more as they move further in generations (Figure 4b). I know some of these points may be beyond the scope of the data to be explained so the authors can perhaps only expand on it in the response and decide if and how they want to add it to the discussion

---

## [Author Response]

Reviewer #1 (Recommendations for the authors):The paper is fairly straightforward to read and beyond a few general comments and methodological questions I have no problem in this being accepted. The authors have done a great job in analyzing a rich dataset to address an important question in the field.Methodology:I did not quite understand how the point-to-point distance for exploration/exploitation was measured (lines 231-233, and lines 110-112 in Supplementary). Were these distances computed as maximum distance between two trajectories within a time window, were they max between trajectories at all time steps? if yes, what time steps. I think a figure in the Supplementary would help.

We did not consider time windows in the computation of the point-to-point distance due to the difficulties inherent in the comparison of time progressions of trajectories from different releases. The point-to-point distance is computed for each point in the focal trajectory and used to label that point. Given a point in the focal trajectory, its distance to the baseline trajectory is defined as the (minimum) distance from that focal point to the closest point within the baseline trajectory. After computing the distances to the baseline trajectory for each point in the focal trajectory, points are labelled according to the set threshold of 300 meters.

In the revised version of the manuscript, we clarified this point in the Materials and methods section “Measuring exploration and exploitation”:

“For each point in a focal trajectory, we determined the point-to-point distance to a baseline trajectory as the minimum distance from the focal point to its closest point in the baseline trajectory. […] Therefore, we used 300 meters as a threshold to group points from a focal route into flight segments differentiated between those exploring new solutions and those exploiting known ones.”

We also clarified the meaning of point-to-point distance in the Results section:

“To do so, we labelled segments of flight trajectories as a function of the point-to-point distance from each point of a focal trajectory to the closest point of the immediately preceding trajectory (i.e., baseline) and compared the exploration–exploitation dynamics both across treatments and between experienced and naïve birds.”

and in the supplementary material:

“We used a distance-based mechanism to label portions of a focal route as either exploration or exploitation depending on the point-to-point distance from each point of the focal route to the closest point of the baseline route at the previous release.”

Discussion:– Can the authors say something about what do these results imply for larger flocks? What would be the limitations of this approach (would the leader bird still be in front for example?)

Although experimental evidence would be necessary to support our speculations, we believe that the dichotomy between leaders in the front of the flock and followers in the back would still hold. Previous experimental studies without generational replacement showed that this is the case even for larger flocks. Nonetheless, the repeated introduction of new naïve individuals, characteristic of the transmission chain design, might introduce more complex turnover dynamics within the flock whose effects are difficult to predict a priori. We revised our manuscript to reflect this discussion and included a new paragraph which also discusses possible connections with the phenomenon of the wisdom of crowds:

“The ability of groups to outperform single individuals by pooling information across their members is an aspect of collective intelligence that has long intrigued researchers. […] However, the repeated introduction of naïve individuals into larger flocks might complicate the dichotomy between leaders and followers by inducing turnover dynamics between the front and the back of the flock.”

– What do the authors think is special about generation 3 that it does not follow the trend in several results?

We believe that the departure of the results of generation 3 with respect to generations 2, 4, and 5 might be due to the relatively small number of samples in terms of pairs per generations/transmission chains (i.e., 10 chains) that were possible to collect during the experiments. With only 10 pairs per generation, it is possible that, by chance, data for generation 3 were more subject to the effects of outliers and limited sample size than other generations. Although the results are significantly different in some cases, these differences are relatively limited while trends are largely in line with those observed for other generations.

– Why do the authors think that the experienced bird starts exploring more as they move further in generations (Figure 4b). I know some of these points may be beyond the scope of the data to be explained so the authors can perhaps only expand on it in the response and decide if and how they want to add it to the discussion

We are not sure how to interpret this comment as both naïve and experienced birds explore less and exploit more as generations progress (red lines in Figure 4b are fitted using pooled data from both naïve and experienced birds). In our answer, we assumed that the reviewer originally meant “exploiting more” instead of “exploring more”. As the reviewer suggested, our data do not allow us to come to a conclusive answer and we thus preferred to elaborate in the discussion. We have now expanded the discussion of this result to further clarify our interpretation:

“Our analysis showed that, in a multi-generational transmission-chain design, the naïve bird has a higher influence than the experienced one during the early generations. […] Over generations, as route information within the pair becomes better, the balance between exploration for route improvement and exploitation of the known route changes in favor of the latter.”

References:

1. Strandburg-Peshkin A, Papageorgiou D, Crofoot MC, Farine DR. Inferring influence and leadership in moving animal groups. Philos Trans R Soc B Biol Sci [Internet]. 2018;373:20170006. Available from: https://royalsocietypublishing.org/doi/10.1098/rstb.2017.0006

2. Galton F. Vox Populi. Nature. 1907;75:450-451.

3. Strandburg-Peshkin A, Farine DR, Couzin ID, Crofoot MC. Shared decision-making drives collective movement in wild baboons. Science (80- ) [Internet]. 2015;348:1358-61. Available from: https://www.sciencemag.org/lookup/doi/10.1126/science.aaa5099

4. Biro D, Sumpter DJT, Meade J, Guilford T. From Compromise to Leadership in Pigeon Homing. Curr Biol. 2006;16:2123-8.

5. Papageorgiou D, Farine DR. Shared decision-making allows subordinates to lead when dominants monopolize resources. Sci Adv [Internet]. 2020;6:eaba5881. Available from: https://advances.sciencemag.org/lookup/doi/10.1126/sciadv.aba5881

6. Stahl J, Tolsma PH, Loonen MJJE, Drent RH. Subordinates explore but dominants profit: resource competition in high Arctic barnacle goose flocks. Anim Behav [Internet]. 2001;61:257-64. Available from: https://linkinghub.elsevier.com/retrieve/pii/S0003347200915641

7. King AJ, Douglas CMS, Huchard E, Isaac NJB, Cowlishaw G. Dominance and Affiliation Mediate Despotism in a Social Primate. Curr Biol [Internet]. Elsevier Ltd; 2008;18:1833-8. Available from: http://dx.doi.org/10.1016/j.cub.2008.10.048

8. Couzin ID, Krause J, James R, Ruxton GD, Franks NR. Collective memory and spatial sorting in animal groups. J Theor Biol. 2002;218:1-11.